# The Politics of Memory: Tradition, Decolonization and Challenging Hindutva, a Reflective Essay

**Bihani Sarkar**

Department of Politics, Philosophy and Religion, Lancaster University, Lancaster, LA1 4YL, UK;
b.sarkar1@lancaster.ac.uk

**Abstract:** This self-reflective essay explores the wider implications of the BJP's inauguration of the Ram Temple in Ayodhya, from the perspective of a scholar of Sanskrit and classical Indian religions. What questions does it raise about our relationship with history, heritage, decolonization and the politics of memory? How can one decolonize oneself and society by reclaiming tradition and heritage, without political agendas and misinterpretations of the past? The article argues for a critical, non-passive, creative, reclamation of tradition for the formation of a truly free decolonized political consciousness.

**Keywords:** Ayodhya Ram Temple; Hindutva; Hindu traditions; Sanskrit; decolonization; faith and heritage; Indian history

## 1. Introduction

'The Night of the sword and bullet was followed by the morning of the chalk and the blackboard'—Ngũgĩ wa Thiong'o

More than fifty years after 1947, India is at a momentous time in its history when its plural democratic identity is being reconfigured before the eyes of its people through unreliable interpretations of a kind of faith. To be Hindu in India nowadays is becoming dangerously synonymous with being a proponent of Hindutva, an essentialization of Hindu traditions that runs counter to its diversity.

On 22 January, over the demolished Babri Mosque in Ayodhya, the BJP government presided over the consecration of a new temple of Ram, symbolic of the monolithic 'Hindu' heritage central to its political messages, the 'master narratives' about which it fabricates.

An openly political act that isolates the temple from the hands of the ascetic Ramanandis, the tradition of Hinduism with centuries of claim to the temple, the foundation has divided Hindus, with one of its highest institutional figures, the Shankaracharya of Jyotir Math in Uttarakhand, condemning the government for aggrandizing it for its own political advantages.[1]

As a scholar of Sanskrit and classical Indian history and religions, I am led to reflect on the role of traditions and heritage in new democracies that are coming to terms with the legacy of colonial rule and who, at the same time, are turning to the wealth of their pre-colonial cultures to rebuild themselves. India's case is an example of political problems that may arise when democracies 'go back' to some sort of past to reclaim an alleged identity—when they are unable to acknowledge social complexities inconvenient for their political ideologies. I hope what I have to say in the following about tradition and decolonization may offer a solution to these problems that isolates neither tradition nor modernity from the difficult, long-term process of decolonization. As will be shown later, my arguments on what decolonization means and how to decolonize are largely inspired by Wa Thiong'o's work on reclaiming tradition, but, in view of misguided attempts such as Hindutva, I also outline important caveats.

Much of this reflection is autobiographical, which may raise a preliminary question—does autobiography count as valid scholarly expression? For years I have been taught to believe that 'autobiographical scholarship' is nothing more than self-indulgence at the expense of truth. As someone trained in a now-dying tradition of holding back the personal voice in scholarly writing, I am uncomfortable with personal admissions, unless necessary. But looking back at the chain of causes and effects that have led me to where I am, I find that everything about my scholarship has been entangled with the personal and the political. I am confronted by the fact that insofar as what I study—India—has shaped, shapes and will continue to shape myself, my personal narrative is a valid and meaningful heuristic device, enabling better knowledge of the things I study and giving me an intimate, intense connection with the world of my subject matter. By studying the Indian past, I consider myself to make an active intervention within a critical moment of Indian history—now—that draws from the passions of my very core: knowing the past is nothing short of knowing myself in the present. As such, scholarly knowing has been for me philosophical understanding, one whose nature is as deeply emotional, connected, and experiential as it is also objective and distanced.

As I hope to show in this essay, scholarly knowing in this personal, connected, emotional, responsive form can also become an act of political intervention, an act of decolonization. Like artistic composition, proper scholarship is about truth-telling, and being committed to the telling of the truth, I find it increasingly important to acknowledge that my own motivations, experiences and emotions are part of the telling. And this acknowledgement of the personal, far from being self-indulgent, has a wider, disruptive purpose: a scholars' emotions can have dangerous uses when instrumentalized with precision and thought. Audrey Lorde speaks of how she learnt to 'train', in other words politically instrumentalize, the anger accumulated from her experiences as a Black woman in 1940s USA 'with accuracy rather than deny this has been one of the major tasks of my life' (Lorde 2007, p. 141). So it is with me. In this essay the autobiographical, the experiential, is instrumental.

## 2. Heritage and Decolonization

Heritage is vital in rebuilding national identities obliterated by colonialism. This is no better exemplified than in the case of Palestine. For modern Palestinians, the celebration of their cultural traditions is a means of resistance: because those traditions are theirs and evoke a memory of a time before Israeli occupation, they are to be fiercely protected and joyfully practiced. For societies transitioning from legacies of European and English governments, heritage represents identity that was free before 'the morning of the chalk and the blackboard', Thiong'o's powerfully succinct phrase for the replacement of African languages by English as the language of prestige in his land.

Yet even as native languages, social organizations, beliefs, cultures, art and practices can give postcolonial democracies their lost voice, it is obvious that there are many aspects of one's heritage that need to be critically re-evaluated to fit modern selves and attitudes. Uncritical efforts to decolonize by a nostalgic return to tradition have contributed to the rise of religious fundamentalism, which sees everything about a 'past' through rose-tinted glasses. This is true of Hindutva.

The desire propelling Hindu fundamentalism in South Asia is to reclaim tradition—a noble enough intention. But its manifestation is violent and divisive. Hindu nationalism, even as it takes pride in a certain sort of heritage, denies that South Asian society was pluralistic, and it misinterprets history to promote cultural division between Hindus and Indian Muslims and Christians.

Without a shadow of doubt, Indian tradition and heritage are diverse and receptive. To repeat certain widely accepted scholarly arguments: Hinduism itself was shaped by external influences as far back as the Vedic and post-Upaniṣadic periods, with interactions described in the Vedas between Vedic speakers and non-Vedic tribes and philosophical arguments in slightly later sources with Buddhist, Jaina and Cārvāka opponents[2] on fundamental beliefs such as the existence or non-existence of a creator and a soul. From

the second century CE onwards, Greek and Iranian religious traditions influenced ideas about Hindu deities, their appearances in art and even the prominence of some of them in the Hindu pantheon.[3] From the 10th century, Islam profoundly changed the landscape of South Asian religious traditions and culture even further. Sulh-i-kull, the Mughal emperor Akbar's visionary policy of universal toleration, underpinned by Sufi ideas[4], was the forerunner of the principle of religious toleration at the heart of the Indian constitution.

The word 'Hindu' is a product of two cultural encounters—Sanskrit speakers with Persians, and Persian speakers with Greeks. An early reference to this encounter appears in Herodotus' *Histories,* in which he describes Greek explorer Skylax's journey down the Indus River in the Sindh region of Pakistan, which the Greeks called Indos. It appears that Skylax had been dispatched on a reconnaissance mission by the Persian emperor Darius, who saw a chance to expand his empire further into the subcontinent. Following or shortly after this expedition, around 515 BCE, Darius conquered the lands around the Indus from which Herodotus said that the emperor received gold-dust as tax–dug up by marvellous ants as large as foxes (Herodotus Book IV. 44 in McCrindle 1901, pp. 4–5).

The original Sanskrit name for the River Indus was Sindhu, but the Persians, unable to pronounce the initial sibilant in the Sanskrit, called the river and everything around that they had annexed into their empire Hindush or Hidus, which became a Persian province, mentioned in Darius' inscriptions in the ancient language of Archaemenid. Herodotus's account was probably based on part-fantastical Persian reports of Hindush. So, the lands of the Sanskrit River Sindhu had become Hindush for the Persians. Herodotus re-translated Hindush into the Greek Indus/Indos, dropping the initial 'h'. Thus in its original meaning, 'Hindu' refers to a geopolitical and even a mythopoeic realm, 'the land where ants dig up gold'—not followers of a religious tradition. Curiously, to be authentically Hindu then would mean to be an inhabitant of Sindh, a Pakistani.

So how can modern India decolonize herself by embracing tradition (and I mean tradition in all its diversity) and making it work for modernity? How can one reclaim oneself while avoiding pitfalls from the past? What is this thoroughly modern old Self?

Much of my own intellectual life, has been spent, mostly unconsciously, in trying to find an answer to these questions, which, I now begin to see, lie at the heart of decolonization.

This life began with my introduction to the Bengali script. I was five years old. I vividly remember my barely contained excitement as I rushed to my grandmother, asking her to introduce me to the alphabet before I started in school. The graceful seduction of those letters had filled my imagination up to that time. The English alphabet which I had learned the year before paled before the mystery and magic of Bengali. For me, knowing the Bengali script meant entering a fairy world, the world of my grandmother's stories. In a way natural for a child, I knew that that world was entirely mine.

This memory stands at the advent of a long road of, not only a fascination with the Indic past, but freeing myself. The road had been first trod by my great-great- and great-grandparents, all anticolonial freedom fighters, and I am more conscious of their passions than ever before. I spent most of my adolescence and my undergraduate immersed in English literature, like so many Indians of my parents' and my generation. Later, I began to study Sanskrit as a second degree. I do not quite remember the complex, emotionally charged chain of motivations that led me to Sanskrit, but somehow it was through my first degree in English and choosing to read Bankim Chandra Chatterjee's *Kapālkuṇḍalā* for a special topics paper—Bankim's Bengali prose makes it difficult for one not to grow curious about Sanskrit—that fired the gun. And formative chats in the book-lined studies of my first Sanskrit teachers at Oxford—James Benson and Alexis Sanderson—helped immensely. By the end of those early exploratory conversations, I made the decision (in the West, mind you!) to tame the beast.

## 3. Ancient Mothers and Decolonization

Then there was Toru Dutt. While reading English at Oxford, I learnt about an ancestor from my grandmother.[5] Toru Dutt was one of the most prolific anglophone writers in the

19th century Bengali literary world, known as 'the mother of Indian poetry in English'. She was also a predecessor of my paternal grandmother, who had been a Dutta before marriage. My grandmother descended from the Hindu line of the Dutt family, which, from 1862, had divided into two lines, Christian and Hindu. Toru's father Govin Chunder Dutt had converted to Christianity in 1862, because of his close ties to the British government. However, both sides of the family regularly interacted with each other (in her letters Toru warmly writes of frequent visits by her Hindu aunts to their Christian house and 'gossip' from the Hindu household) even though each side was very different. The Christian Dutts worked for the British civil service, travelled frequently to England and France (Toru attended lectures in Cambridge) and the entire household wrote, read and immersed themselves in French and English literature. From her homes in Rambagan and Bagmari near Calcutta, Toru wrote two volumes of poetry, one of which was a translation into English metrical verse of nearly two hundred French ballads, another of Hindu myths from the *Viṣṇupurāṇa*, a French novel and sixty letters to her friend Mary Martin. Encouraged to be polymaths and to travel outside India by a loving father ahead of his times, Toru, along with her sister Aru, fused different languages and worlds, Francophone, Anglophone, Sanskrit, Bengali, Hindu, Christian, Calcutta, into their fertile imaginations, with both play and discipline.

My grandmother was proud of Toru. She had spoken about her courage to me with warm sentiment, even though Toru was from the dissident side of the family. Toru was not an intellectual project for her, as she may be for a professional academic. Toru was real. My grandmother was passing down to me her memory to keep alive.

In 2004, I decided to write a thesis on Toru Dutt as an optional ninth paper in addition to the usual eight papers for the Final Honours School in English Language and Literature. I started reading Toru's letters to Mary Martin, who had become her close friend during a visit to Cambridge. It was in finding Toru's voice in those letters, sounding through quotidian narratives of a time and world gone by, that I connected intensely with what naturally and inexplicably felt to be a memory of one of my own, revivified in the act of reading. It felt that Toru, one of my ancient mothers, was speaking to me.

By 1876 Toru was evidently dying. She reports coughing up blood in her letters to her friend Mary Martin, and by 1877 she had died of tuberculosis-- at the age of 21.

But before she died, Toru had embarked upon a last linguistic project: learning Sanskrit. In her letter to Mary Martin dated November 23rd, 1875 Toru writes:

'Papa [Govin Chunder Dutt] and I are going to learn Sanskrit in December; Papa says as there is no good opportunity to learn German now, we had better take up Sanskrit instead of doing nothing. I am very glad of this. I should so like to read the glorious epics, the *Ramayana* and the *Mahabharata,* in the original. I shall be quite a Sanskrit pundit, when I revisit old Cambridge'. (Das 1921, p. 112)

By 13th January 1876 she and her father were completing Sanskrit grammar and looking forward to reading the *Rāmāyaṇa*:

'My dear Mary….We are going on with our Sanskrit lessons now. When we have finished the book we are reading now, we shall take up Valmiki's *Ramayana*. My uncle [either Har or Girish one of Govin Chunder Dutt's two brothers] has followed our example, and has commenced reading Sanskrit also, with another pundit'. (Das 1921, p. 125)

And, indeed, by May 13th 1876, she had started the Sanskrit epics: 'Our Sanskrit is going on but slowly. We are now reading extracts of the *Mahābhārata*' (Das 1921, p. 159).

After reading about Toru's unfinished work, my own intention to learn Sanskrit crystallized from various inchoate passions into something familial, simple, and visceral. Blood is a powerful bond. Remembering Toru and seeing that her final great endeavour lay unfinished were the decisive reasons for my learning Sanskrit. I was reclaiming memory, and lost self, like Toru more than a century ago.

### 4. The Decolonized Subject

In *Decolonizing the Mind*, Ngũgĩ wa Thiong'o had argued that to free postcolonial identity of the inner and imaginative oppression of colonial rule, it was necessary to go back to one's own language(s). This is 'decolonizing the mind'. 'Unfortunately, writers who should have been mapping paths out of that linguistic encirclement of their continent also came to be defined and to define themselves in terms of the languages of imperialist imposition. Even at their most radical and pre-African position in their sentiments and articulations of problems they still took it as axiomatic that the renaissance of African cultures lay in the languages of Europe. I should know,' he ends wryly. Wa Thiong'o was a prolific writer in English.

From 1977 he wrote only in Gikuyu, his mother tongue (Wa Thiong'o 1986).

Wa Thiong'o's idea can be pushed further. Decolonization need not be a process simply related to language. It can mean a reclamation of the full wealth of 'what was ours', including that which has become lost or lies in the margins, that weaves the totality of the individual self. It means looking back, looking into corners, finding what was there, perhaps being surprised, treasuring what we find, but also having the freedom to play with what is found, creatively, sometimes in the spirit of love, sometimes in the spirit of rejection. Key to this process of loving and critical play with the past is the knowledge that 'that past is mine and I am free to do with it as I like'.

A democracy shaped by such a decolonized subject can form the ground for a new kind of creative political consciousness, who sees inheritance clearly in all its diversity, historical complexities and paradoxes and embraces it with pride and love, but is also liberated from its expectations—a subject who is at once naïve and rebellious. In one sense, this decolonized subject is akin to a Hindusthani classical musician, who, steeped in his/her tradition, can forge an intense visionary connection with traditional forms of *rāga*s, refined by both Muslim and Hindu masters, filled with songs to both Hindu gods and Allah. However, in full spate of creative play—what the 9th century Sanskrit aesthetician Rājaśekhara calls the shining of *pratibhā*, the light of imagination—s/he can alter and interpret its form to make something completely new.

Indeed, had she survived and continued Sanskrit and translation, Toru could have become the perfect exemplar for such a decolonized subject and 'new India'. I would like to think that a character of her independent, romantic, kind nature in whom incipient feelings of nationalism were already rising (as seen in her letters) would surely have joined the Indian national movement with others of her generation. At ease with the cosmopolitanism of the mixed worlds she inhabited, questioning in her regard of tradition, she could have yet been fiercely passionate and proud of her native core and heritage, a critic of colonial impositions.

### 5. Scholarship and Creativity; Subservience and Rebellion

Only recently did I realize that my own intellectual journey shares a parallel with wa Thiong'o's awakening. Like wa Thiong'o, I became conscious that two hundred years of English education had led to my being mainly English-speaking and knowing mainly Western intellectual canons. So, I learned Sanskrit to divest myself of English, to understand the heritage and traditions that my grandfather, great-grandfather, great-grandmother and great-great-grandmother, freedom fighters in the Indian war of independence, fought to reclaim. By learning 'Sanskrit', I was learning not just a language but recognizing and embracing a behemoth comprising religion, literature, culture, aesthetics and society.

I wanted direct access to the behemoth.

This language opened to me a splendidly arrayed world of the symbols, sounds, thoughts and lives of my people; of the accumulated traces of memories, what wa Thiong'o calls 'the collective memory bank of a people's experience in history' (Ibid.); or what the Buddhists call *vāsanā*s, latent imprints of past lives in layers of human consciousness. Reading Sanskrit allows me to commune and connect intensely with shadowy *vāsanā*s, human lives and voices calling out from a time once lived. A deep ancient powerful

memory comes alive in a moment of intense recognition. It is a memory at once familiar and also seductively unfamiliar and alien.

But this great love was not the only reason for claiming—or taming—Sanskrit, because as a woman, a bystander to a tradition benefiting men, I was critical, unlike wa Thiong'o, whose adoration of African traditions was unalloyed. Combined with the need to return to one's cultural womb was a visceral reaction against voices of authority and oppression, who mediated that Sanskritic tradition to me. The language, history and traditions I loved were/are profoundly patriarchal—in their most Brahmanical, orthodox expressions. For example, the wearing of the sacred thread, the rite of passage in Brahmanism initiating one into the study of the sacred texts, the Vedas and access to knowledge, is permitted only to men among the 'twice-born' (*dvija*) group (the three upper *varṇa*s born once through the womb and a second time through the Vedas) and not to women.[6] Within the ritual stages and processes ordering life, women cannot be born a second time through the Vedas, as men are, because they have no formal initiation into the scriptural traditions of Brahmanism. And while women interlocutors appear in the Upaniṣads as audiences for their husbands' sagacious sayings, in general the orthodox Brahmanical view, as expressed in the *Lawbook of Yājñavalkya*, is that men from the three upper *varṇa*s carry forward the work of their fathers, elevating their ancestors' progressive ascent in the afterworld through their commitment to dharmic acts: 'through a son, a grandson, and a great grandson a man attains worlds, eternity and heaven'.[7] 'Therefore' [the Hindu lawbook continues] 'women are to be honoured and kept well-protected' since they are secondary instruments to the exaltation of lineage, and not active agents in the process.

Much of this attitude towards women in Hindu law is shared by the ancient ascetic traditions of India, Brahmanism, Buddhism and Jainism, in which women came to be especially objectified, reviled and feared, because they were seen as seductive obstacles to the ascetic ideals of restraint (*yati*) and abnegation. Women and their bodies, for the men engaged in world-denying practices, are simply sense-exciters, perturbing the male hermit withdrawing from the world in intense self-contemplation.

And when it came to the female renunciate, the *bhikṣuṇī*, her autonomy was treated with ambivalence. The figure of the independent female hermit who can travel from place to place at will is deemed suspicious in legal discourse in *Manu*, the *Arthaśāstra* and the *Kāmasūtra*, stereotyped as either the spy or the 'cunning agent encouraging illicit sexual behavior in others' (Jamison 2006, p. 23).

On the other hand, within Brahmanism itself, descriptions of the female can also exuberantly embrace her autonomy and wildness, vital aspects of power. Poems to the warrior goddess Durgā exult in her single combat with a buffalo demon, whose blood she wears as lac on her foot. They rhapsodize her 'unfeminine' love for the cremation ground, whose skulls are like exquisite geese flocking to her (according to the Prakrit poem the *Gaüḍavaho)*. And just as they choose to regard her connections to the macabre as poetic, they delight in rather than treat suspiciously her evident womanly sexuality: her voluptuous body, her glittering jewels that charm devotees, the coquettish jingle of her anklets and her passion for her consort Śiva. Myths about the goddess underline her power by emphasizing that she achieved alone what the other gods could not achieve together. Here was a perception that a woman bettered the man, and even as she challenged the norm, she was not 'on the outside' but at the center of the order of power and politics. The Goddess (Śākta) traditions are the single most important cultural expressions of women's experiences, bodies and capacities beyond those that appear from the normative Brahmanical pale.

Then there was the view in aesthetics that poetry—the highest kind of speech and the most respected form of learning—was like a woman in providing instruction while giving delight draped in ornaments (*alaṃkāra*, the word for ornament, in the context of poetics meant figures of speech). There was an equation of the sensory, almost tactile world of Sanskrit poetry with the equally sensual pleasures of *rasa* embodied by the woman. This may be seen as a kind of misogyny: that women give pleasure rather than take pleasure, that they are instruments rather than agents. On the other hand, one can also say that in the

Indian poetic culture, silence did not equate to the female; rather, knowledge and speech did. It is significant that the point of comparison is Poetry—words, rhetoric.

Moreover, a range of perspectives on women 'behaving badly' appear, from early Sanskrit love poetry on women seeking and expressing love in defiance of social norms and married women enticing travelling men (these in collections of *subhāṣita*s, 'fine sayings') to a goddess pursuing pious austerity on her own for self-empowerment and ultimately love (this in the epic poem the *Kumārasambhava*).

We find that there was no single view on what counted as female transgression and how women should behave but that attitudes in South Asia were often paradoxical, some celebrating women's agency,some denouncing it, some falling in between and all co-existing in dialogue. Ramanujan famously spoke about these as 'counter-systems' in women's tales (Ramanujan 1991).

Clearly, Hindu traditions beyond religious rules offer a wider spectrum of attitudes to women. But those alternative voices and their importance for today in widening the horizons of what counts as a Hindu womanhood, of breaking strict binaries of right and wrong, are hardly spoken about by mainstream Hindu culture. The proponents of orthodox Hinduism, and undoubtedly the proponents of Hindutva, would assuredly deny their existence.

Considering my own experiences, which are valid and meaningful, and these 'counter-systems' of womanhood, I am strengthened in my conviction that the oppressions of a normative past have not and can never work for me as a modern Indian woman.

'Hindu' feminism as an intellectual and active effort would therefore involve continuously pointing out counter-normative traditions and metaphors, including and *especially* the Śākta, and fighting for equal authority in learning, interpreting and teaching scriptural knowledge and performing worship. Alongside men, there should be householder women as priests and teachers. Modern Hindu movements such as the Sarada Math mission, in which women can become respected religious leaders, act as priests and climb the institutional hierarchy, are few and far between. Moreover, in the case of Hindu women's reform movements, women need to leave the world in order to attain religious power and prestige—the substance of power may be attained only by being outside society. To have a large-scale movement involving ordinary householder women in society would require a fundamental challenge to the idea of Hindu ritual purity, which views all women as less pure than men, and thus to Brahmanism itself, grounded on the sacrosanct status of ritual purity. In this sense, the object of 'Hindu' feminism would be exactly that of Ambedkar's challenge to caste: it would require dismantling caste altogether, to reconceptualize and uplift the personhood of women from the notion of purity. Hindutva's own idea of glorious womanhood is Sītā, the suffering, devoted and noble wife of Ram, its chief heroic figure. Goddesses, dismantlers of ritual purity and yet loved by so many, and women religious figures who behave atypically and question norms, yet remain respected social actors, do not form parts of their messages.

Curiously, Sanskrit was both the Great Love and the Enemy to be battled. My motivations to learn it were both nostalgic and rebellious. Who says decolonization is straightforward!

As I tell my students, decolonization is a twinned process of being both loving and subversive, and we must play a balancing act between knowing well and challenging to arrive at something that works for us. A deep passionate love for traditions marks the process removing traces of colonial subjugation that wa Thiong'o appealed for in the case of African civilization. With this love comes work—a disciplined practice of engaging with tradition. One can cherish and preserve tradition only by knowing and practicing it, but, more empoweringly, knowledge enables one to critique properly and innovate (as the structure of traditional Indian philosophical argument shows us). And one must be ready for surprises, to discover things one didn't know about, which may alter current assumptions about the past.

With knowledge of tradition comes the more arduous task of thinking about how that past relates to our present identities, and how, if it does not serve us, it will need change.

An opponent might argue that my interest in tradition goes hand in hand with fundamentalist beliefs. But nothing could be further from the truth: Hindutva's harshly puritanical, narrowly parochial and imaginatively impoverished interpretation of Hinduism could not be farther from the poetic, mystical, sensuous, generously expansive forms of faith dear to my heart, and which are native so many parts of the subcontinent, as it is to the most ancient texts of Hinduism. Outlier figures, Muruggan, Bhairava, Rudra, the Śabara, *baba*s, *fakir*s and *mataji*s of no particular ilk, but above all, the Goddess and goddesses, have always peopled the canvas of the traditions I love and their narratives and voices are luminous threads binding Hinduism's tapestry.

## 6. Modernity and Tradition

This brings me to my final point: the ideological tension between modernity and tradition in Indian society at present. Hindu nationalists present themselves as champions of an exclusionary Indian tradition because advocating for a Hindu 'democracy' serves their interest to be seen as preservers and protectors of cultural resources—as guardians of decolonizing India from centuries of traumatic 'Western' influence. In reaction, liberal Indian academia voices an essential challenge to this narrow view of cultural heritage and upholds academic freedom at a time when it is being suppressed. However, liberal academics sometimes regard any study of the pre-colonial Indian past as serving a Hindu fundamentalist's conception of India and many social aspects of pre-colonial culture through stereotypes (for example, 'Sanskrit is the language of elite control' or 'All South Asian women were oppressed and passive').

In fact, Westernized liberal academics approach Indic heritage (including the interpretation of early Indian texts) through the values of liberalism, which sees tradition, religious, and cultural, as oppressive and backward, or ignores tradition completely to find all answers in a 'progressive' West. This is the supercilious embarrassed attitude that underlies the liberal's horror at the hijab, even when thousands of Muslim women may choose to wear it proudly as an expression of their beloved faith. It also underlies the pernicious stereotypes in the Western media, particularly after 9/11, about the 'powerless Muslim woman'.[8] Behind the liberal's aversion to any expression of faith and spirituality lies an intolerance and elitism the same as the Hindu fundamentalists' 'consciousness of kind'. It is also one and the same as the embarrassment of the anglicized Indian at his/her own ethnicity, with the non-Western, dark 'Other' with which, though a part of his/her identity, s/he is always uncomfortable.

Indian democracy is divided into two ideological camps: Western liberalism and Hindu fundamentalism. While liberals associate themselves with progress, secularism, 'Westernization', individual rights, enlightenment, the Hindu fundamentalists with collectivism, faith and spirituality, the past, ancient languages, classical culture and dangerously, native-identity (versus globalized Western culture). This has led to a Huntingtonian 'clash of civilizations' between the camps, with those who love and celebrate tradition, but do not share any of the values and ethics of Hindu fundamentalism, nor the Eurocentrism of liberalism, increasingly isolated. And this divide does not serve in decolonizing Indian democracy, with liberals unwilling to engage with tradition in a truly sympathetic way and Hindu fundamentalists interpreting heritage and history in a narrow way to exclude Islam and pigeon-hole religious identity.

The older middle ground of liberal Hinduism, which many modern Hindus have grown up in since Independence, seems be dying a gradual death in this present schism. Liberal Hindus lived and practiced their faiths in an easy, indefinable and often pantheistic way. They did not determine their identities in terms of 'religion'— identities were formed by allegiances to languages—and often aspects of many other religious traditions, Buddhist, Jain, worship of outlier goddesses such as Kālī and her forms, and Muslim, would have intersected unquestioningly in their worship—this even for strict brahmans. But since its formulation in 1923, Hindutva, manipulating collective cultural feelings of loss and yearning for heritage and identity, demands that Hindus circumscribe themselves with

an idea—politically imposed from above and alien to many. Moreover, it is profoundly uninterested in either change or creativity among Hindus. Nostalgia in Hindutva does not mean originality and artistry.

## 7. Conclusions: Looking Ahead

By proclaiming itself the most authoritative custodian of Indian heritage, the BJP is hijacking decolonization, so that to decolonize now means to 'go Hindutva'.

It is more important than ever before to connect tradition with modernity for Indian democracy to move further in its postcolonial journey. It is more important than ever before to reclaim pre-modern South Asian languages and traditions, living and dead, to challenge the BJP's black and white stories of the past with nuanced, thoughtful stories and creative, new, rule-breaking articulations of heritage. It is more important than ever for modern open-minded questioning Indians to learn Sanskrit, the very queen of all the ancient Indian languages before it is completely appropriated in the Hindutva- project and becomes the bugle of Ayodhya. I call this a creative, critical, playful and non-passive reclamation.

I appeal for a reclamation of tradition while being a modern South Asian, who has lived in different places, mixed with different people, broken rules, spoken some Western languages, lived with contradictions and many selves, has conflicts with the past.

For Dr. Julie Hearn, who introduced me to Ngũgĩ wa Thiong'o and, most importantly, taught me courage, and the first students of my course 'Religion and Politics in South Asia: the Power of the Past' from whom I learnt so much. Thank you.

**Funding:** This research received no external funding.

**Institutional Review Board Statement:** Not applicable.

**Informed Consent Statement:** Not applicable.

**Data Availability Statement:** No new data were created or analyzed in this study. Data sharing is not applicable to this article.

**Conflicts of Interest:** The author declares no conflict of interest.

## Notes

1  https://theprint.in/india/shankaracharya-not-to-attend-temple-consecration-says-not-anti-modi-but-not-anti-dharma-either/1917780/ (accessed on 15 March 2024).
2  See for example Thapar (2021).
3  See for example Srinivasan (1997).
4  See for example Moin (2022).
5  In fact this dissertation on Toru Dutt developed into an article, published in 2007: Sarkar (2007).
6  See for example the *Yājñavalkyasmṛti* 1.10-50 in (Goodall 1996, pp. 295–301).
7  *Yājñavalkyasmṛti* 1.78 in (Goodall 1996, p. 305).
8  See for example Abu-Lughod (2013) and Mahmood (2006).

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
