# Peer review of "The Politics of Memory: Tradition, Decolonization and Challenging Hindutva, a Reflective Essay"

_religions, doi:10.3390/rel15050564_

Round 1

Reviewer 1 Report

Comments and Suggestions for Authors

This is a wonderful little essay, filled with great insights. I recommend publishing it, with minor revisions. The suggestions below are mainly to make the arguments clearer and expand some of the points. I’m assuming that since this is a short essay, the author will have the space to do so.

1.       I love the opening quote (‘The Night of the sword and bullet was followed by the morning of the chalk and the 15 blackboard’—Ngugi wa Thiongo) but the writer should allude to its relevance to the paper sooner. A sentence like the following one could appear earlier: “…my arguments on what decolonization means and how to decolonize are largely inspired by Wa Thiongo’s work on reclaiming tradition, but, in view of contemporary bigoted and misguided attempts such as Hindutva…”

2.       Lay out the main argument about tradition and modernity sooner. And also tell the reader sooner why the author decides to write about their own experiences.

3.       Please expand on this fascinating and important paragraph. Most readers will not know what the author describes here:

“An openly political act that isolates the temple from the hands of the ascetic Ramanandis, the tradition of Hinduism with centuries of claim to the temple, the foundation has divided Hindus, one of its highest institutional figures the Shankaracharya of Jyotir Math in Uttarakhand, condemning the government for aggrandizing it for its own political advantages.

What did the Shankaracharya say? How deep do these divisions run?

4.       The point about Palestine and the celebration of cultural traditions as a form of resistance is really interesting. The author might note that Hindutva forces are seeking to appropriate anti-colonial nationalism by claiming that they are engaging in a project of decolonization.

5.       The sentences from 166 to 202 are original and excellent. I’d love for the author to elaborate on them. I’d love to know whether feminist Sanskrit scholars have engaged in the playful, creative, expansive revisionist analysis that the author invites.

6.       Conclusion: I’m left wondering why scholars and activists have not engaged in more of a creative reclaiming of Hinduism/tradition and whether the author might speculate on this question. I don’t think this can simply be explained by the intelligentsia’s liberal-secular bent. After all, post-colonial scholarship has engaged in a fierce critique of both liberalism and secularism.

Author Response

Thank you for your comments.

Reviewer 2 Report

Comments and Suggestions for Authors

This is an interesting, important and urgent article. It lays out in some detail the faultlines behind the proliferation of modern Hindutva and the knotty issues surrounding the place of modernity and tradition in defining Indian identity. While a short reflective piece may not address all issues, it needs closer thinking on the following points:

1. It is great that the author draws on Ngugi wa Thiongo and Audrey Lorde as an invitation for self-reflection. However, we get remarkably little sense of the author's social location beyond her gender and that she was born in a family of Indian nationalists. This 'locating' is important as it has some bearing on the nature of decolonisation proposed. It might be wroth noting that Lorde was writing as an African American lesbian feminist, a position of extreme marginality on most counts. 

2. The broader point that one needs to critically engage with the inheritance and language(s) of Sanskrit is well taken. Yet, no reference is made to the many non-Sanskrit languages and traditions that have existed in the ancient Indian subcontinent, before, during and in parallel. Of particular concern is the absence of any mention to the equally flourishing Dravidian languages or Tamil classical traditions, or those of Sino-Tibet, not to mention the many indigenous/Adivasi cosmologies that dot India. Again, this has bearing on the nature of decolonisation argued for. 

3. This brings me to the third point. The piece does not offer a clear account of what decolonisation would actually mean in practice. Yes, it does mean embracing one's tradition in its full range but this raises more questions than it answers? How does one define one's tradition? Why is this to be found only through reading Sanskrit?  Decolonisation comes across as mostly an intellectual process to be rendered through a reading of texts and discourse. How does it relate to actual practice? It might be worth reflecting  on what this might mean for social minorities like Dalits and Adivasis. The author might be aware of a troubling strain in Indian sociology, represented by the likes of GS Ghurye, AR Desai and RK Mukherjee (all high caste Hindus, and fluent in English), that British colonialism interrupted a long duree process of sanskritisation that saw the gradual incorporation of outcaste and tribal communities in the grand fold of Hinduism. They argued that postcolonial India should resume that process, a project that RSS took up quite wholeheartedly.  

4. More reflection is required on the nature of ideological divisions between Liberals and Hindu nationalists. The social class that predominantly constitutes the leadership of both types are not radically different: they tend to hail from largely upper caste/class Hindu backgrounds. Many if not all Hindu nationalist leaders, until very recently, were well versed in English education. This suggests a shared encounter with forms of Western education but with markedly different consequences. Again, this might be helpful to reflect on the internal contradictions in both liberal secular and Hindu nationalist projects. Both engage with forms of western modernity but neither can really go back to an ideal state they imagine as desirable. 

Author Response

Thank you for your comments.

Reviewer 3 Report

Comments and Suggestions for Authors

This is an exceptionally strong and important personal essay that is well-informed both by the author's scholarship and her direct experiences with the cultural and political discourses with which she is engaging. It also reflects a terribly important perspective distinct from the polarized positions that currently dominate India's political discourse. This is the perspective of those, in the author's words "who love and celebrate tradition, but who do not share any of the values and ethics of Hindu fundamentalism, nor the Eurocentrism of liberalism." As, I must confess, a fellow member of this 'third camp,' I am deeply in sympathy with this essay and I believe it represents a voice that needs to be heard at this time. I strongly favor its publication. 

Author Response

Thank you for your comments.